# Endoplasmic Reticulum Protein Disulfide Isomerase Shapes T Cell Efficacy for Adoptive Cellular Therapy of Tumors

**DOI:** 10.3390/cells8121514

**Published:** 2019-11-26

**Authors:** Katie E. Hurst, Kiley A. Lawrence, Lety Reyes Angeles, Zhiwei Ye, Jie Zhang, Danyelle M. Townsend, Nathan Dolloff, Jessica E. Thaxton

**Affiliations:** 1Department of Orthopedics, Medical University of South Carolina, Charleston, SC 29425, USA; hurstk@musc.edu (K.E.H.); lawrenki@musc.edu (K.A.L.); 2Department of Cell and Molecular Pharmacology and Experimental Therapeutics, Medical University of South Carolina, Charleston, SC 29425, USA; reyesl@musc.edu (L.R.A.); tezzh@musc.edu (Z.Y.); zhajie@musc.edu (J.Z.); dolloffn@musc.edu (N.D.); 3Department of Drug Discovery and Biomedical Sciences, Medical University of South Carolina, Charleston, SC 29425, USA; townsed@musc.edu; 4Department of Microbiology & Immunology, Medical University of South Carolina, Charleston, SC 29425, USA; 5Hollings Cancer Center, Medical University of South Carolina, Charleston, SC 29425, USA

**Keywords:** protein disulfide isomerase, unfolded protein response, ER stress, T cell, tumor cell, ubiquitin, immunotherapy, redox

## Abstract

Effective cancer therapies simultaneously restrict tumor cell growth and improve anti-tumor immune responses. Targeting redox-dependent protein folding enzymes within the endoplasmic reticulum (ER) is an alternative approach to activation of the unfolded protein response (UPR) and a novel therapeutic platform to induce malignant cell death. E64FC26 is a recently identified protein disulfide isomerase (PDI) inhibitor that activates the UPR, oxidative stress, and apoptosis in tumor cells, but not normal cell types. Given that targeting cellular redox homeostasis is a strategy to augment T cell tumor control, we tested the effect of E64FC26 on healthy and oncogenic T cells. In stark contrast to the pro-UPR and pro-death effects we observed in malignant T cells, we found that E64FC26 improved viability and limited the UPR in healthy T cells. E64FC26 treatment also diminished oxidative stress and decreased global PDI expression in normal T cells. Oxidative stress and cell death are limited in memory T cells and we found that PDI inhibition promoted memory traits and reshaped T cell metabolism. Using adoptive transfer of tumor antigen-specific CD8 T cells, we demonstrate that T cells activated and expanded in the presence of E64FC26 control tumor growth better than vehicle-matched controls. Our data indicate that PDI inhibitors are a new class of drug that may dually inhibit tumor cell growth and improve T cell tumor control.

## 1. Introduction

Recent advances in cancer immunotherapy have given rise to new therapeutic classes that dually target tumor cells directly and promote anti-tumor immunity by modulating T cell function [1,2]. Targeting the endoplasmic reticulum (ER) stress response is a strategy to induce a pro-death response in tumor cells that leads to control of tumors [3,4]. For example, gene deletion of the protein kinase R (PKR)-like ER kinase (PERK) target activating transcription factor 4 (ATF4) inhibits pro-survival adaptations to cell stress and leads to regression of tumor growth [5]. In contrast, using T cell-specific conditional gene deletion of PERK, we demonstrated that PERK restricts T cell anti-tumor capability [6]. Together, these data suggest that targeting the unfolded protein response (UPR) may be a therapeutic avenue to develop effective new therapies for cancer patients. Leveraging the UPR-mediated cell death pathway led to the development of a new class of therapeutics that target protein disulfide isomerase (PDI), a redox-dependent protein folding enzyme with isomerase and chaperone activity. In multiple myeloma cells, inhibition of PDI led to an accumulation of unfolded/misfolded proteins demonstrated by increased ubiquitination, rapid cell death through activation of the UPR, and enhanced efficacy of FDA-approved proteasome inhibitors [7,8]. PDI is an emerging drug target in oncology, and, while the anti-tumor effects of PDI inhibition have been well documented, the consequence of PDI modulation on healthy immune cells has not been assessed. 

PDI mediates protein homeostasis (proteostasis) through facilitating protein folding, and plays a less defined role in signaling [9]. The PDI family is comprised of over 20 oxidoreductase enzymes containing thioredoxin folds, allowing disulfide bond formation between cysteine residues of nascent polypeptide chains. Specifically, PDI is a dithiol–disulfide oxidoreductase chaperone that induces stability in native protein folding through catalysis of cycles of thiol oxidation and reduction [9,10]. PDI plays an opposing role in cell survival and apoptosis following ER stress, depending upon its redox state [11,12]. Redox-dependent proteins are proving to be effective drug targets in a wide variety of pathologies, including cancer [13]. More recently, these efforts encompass modulating host immunity. T cells with an abundance of cell surface free thiols [14] or overexpression of thioredoxin [15] improve their capacity to control tumor growth. 

The redox state of PDIs catalytic cysteine residues is maintained through protein–protein interactions with ER oxidoreductase 1 alpha (ERO1α). ERO1α preferentially reacts with reduced PDI, leading to oxidation of PDI dithiols to disulfides which can then serve as electron acceptors in the folding of nascent proteins [10]. A consequence of electron transfer from ERO1α to molecular oxygen is the generation of hydrogen peroxide [16,17]. This can lead to oxidative stress and activation of the UPR [18]. To maintain homeostasis, the oxidizing environment of the ER lumen is buffered by the ratio of reduced to oxidized glutathione, GSH-GSSG, respectively [19]. Persistent ER stress due to a chronic load of unfolded/misfolded proteins in the ER lumen activates the PERK/ATF4 pathway aligned with the transcription factor C/EBP homologous protein (CHOP) that induces generation of reactive oxygen species (ROS) and cell death through ERO1α-inositol triphosphate receptor (IP_3_R)-Ca^2+^ pathway [20]. In T cells, we have demonstrated that genetic deletion of PERK diminished ROS generation in the mitochondria and improved T cell-mediated tumor control [6]. We also found that pharmacological inhibition of ERO1α or IP_3_R reduced the burden of mitochondrial ROS in T cells and promoted mitochondrial ATP production, leading to better T cell tumor control [6,21]. Effector cells are a secretory T cell subset committed to new protein synthesis. In contrast, memory T cells are a quiescent subset not disposed to high levels of protein translation [22]. Our recent work suggested that effector T cells may be subject to the terminal UPR [6], but assessment of unfolded/misfolded proteins has not been addressed. 

Here we assessed the effect of pan-PDI inhibitor E64FC26 [8] on healthy Pmel TCR-transgenic CD8 T cells specific for gp100 melanoma antigen in vitro and in vivo. We demonstrate the surprising finding that PDI inhibition attenuates the UPR and promotes viability of healthy T cells, whereas it activates the UPR and restricts survival in oncogenic T cells. We show that limited UPR activation and PDI expression is a feature of memory-like T cells that is recapitulated in PDI inhibitor-treated T cells. We demonstrate for the first time that effector T cells are predisposed to a terminal UPR. In line with these data, PDI inhibitor-treated T cells express memory T cell traits associated with stemness and anti-tumor metabolism. Thus, we find that T cells conditioned ex vivo with E64FC26 over the course of activation and expansion promote superior tumor control. These data suggest that PDI inhibitors are a novel strategy to develop for treatment of cancer patients, that concurrently induce tumor cell death while nourishing anti-tumor immunity. 

## 2. Materials and Methods

### 2.1. Mice

Pmel (B6.CgThy1^a^/CyTg(TcraTcrb)8Rest/J) and C57BL/6J mice were obtained from the Jackson Laboratory. All animal experiments were approved by the Medical University of South Carolina (MUSC) Institutional Animal Care and Use Committee (IACUC) for protocol IACUC-2018-004422 (Approval date: 24 October 2018) under Animal Welfare Assurances number A3428-01. The Division of Laboratory Animal Resources at MUSC maintained all mice. 

### 2.2. T Cell and Tumor Cell Culture

For T cell cultures, whole splenocytes from Pmel mice were activated with hgp100 peptide (GenScript, Piscataway, NJ, USA) and expanded with 200 U rhIL-2 (NCI, Frederick, MD, USA). For cytokine differentiation, after 3 days of expansion T cells were split into media containing 200 U rhIL-2 or 50 ng/mL rhIL-15 (Shenandoah, Warwick, PA, USA) and harvested for analysis. For drug treatments, whole splenocytes from Pmel mice were activated and expanded in the presence of vehicle, 0.5 μM E64FC26, or 5 μM EN460 (MP Biomedicals, Irvine, CA, USA) for 3 days. Thereafter, T cells were split into new media, rhIL-2, and fresh drug was added for 4 more days of expansion. B16F1 tumor cells (ATCC, Manassas, VA, USA) were maintained in RMPI complete T cell media and passaged three times prior to in vivo inoculations. Tumor cells were determined to be mycoplasma free in November 2019. All medias were supplemented with Mycoplasma prophylactic (Invivogen, San Diego, CA, USA). Hut78 and Jurkat T cells were maintained in RPMI and treated with 0.5 μM E64FC26 for 16 h prior to cell harvest. An automatic cell counter was used to quantify live cells at points of harvest. 

### 2.3. Immunoblotting

T cells were lysed in RIPA buffer (Sigma, St. Louis, MO, USA) supplemented with protease inhibitor cocktail (Cell Signaling Technology, Danvers, MA, USA) and phosphatase inhibitors I and II (Sigma, St. Louis, MO, USA). Protein concentrations were normalized using Pierce BCA Kit (Thermo Fisher, Waltham, MA, USA) and loaded to 4%–10% agarose gels (BioRad, Hercules, CA, USA). Primary antibodies for Ubiquitin, ATF4, TCF7, HK2, β-actin, Tubulin, and anti-rabbit secondary were obtained from Cell Signaling Technologies, Danvers, MA, USA. p-ACC was obtained from Thermo Fisher (Waltham, MA, USA).

### 2.4. Redox State Determination of Ero1α and PDI

The oxidation state of PDI and Ero1α were assessed in cell lysates under both nonreducing and reducing conditions using 2-Mercaptoethanol (Calbiochem, Danvers, MA, USA) SDS-page of stain-free TGX gel (BioRad, Hercules, CA, USA) followed by immunoblotting with Ero1α (Santa Cruz Biotechnology Inc., Dallas, TX, USA) and PDI antibodies (Cell Signaling Technology, Danvers, MA, USA).

### 2.5. RNA Analysis

RNA was isolated with RNeasy Mini Kit (QIAGEN, Hilden, Germany) and single-strand cDNA was made with High Capacity RNA-to-cDNA Kit (Applied Biosystems, Foster City, CA, USA). UPR gene arrays were purchased from (Thermo Fisher, Waltham, MA, USA) and run according to manufacturers’ protocol. Human and mouse Taqman gene probes (Applied Biosystems, Foster City, CA, USA) were used to perform real-time PCR using the StepOnePlus Real-Time PCR system (Applied Biosystems, Foster City, CA, USA). Gene expression for ATF4 *(Atf4)*, CHOP *(Ddit3)*, ERO1α *(Ero1l)*, Cpt1a (*Cpt1a*), HK2 (*Hk2*), and TCF7 (*Tcf7*) were normalized to GAPDH (*Gapdh*).

### 2.6. FACS Staining and Analysis

Fluorochrome-conjugated monoclonal antibodies CD62L-FITC (MEL-14), CD44-PE-Cy7 (IM7), and CD8-APC (53-6.7) and respective isotype controls were purchased from Thermo Fisher (Waltham, MA, USA). Extracellular stains were performed in PBS supplemented with 5% FBS. Mitochondrial ROS or global peroxyl, hydroxyl, and other cellular ROS were measured with MitoSOX Red Mitochondrial Superoxide Indicator (3 μM, Molecular Probes, Eugene, OR, USA) and DCFDA/H2DCFDA probe (1 μM, Molecular Probes, Eugene, OR, USA), respectively, loaded at 37 °C for 30 min in RT PBS. Extracellular stains were added post dye incubation. For Annexin staining, cells were washed and stained using Annexin V-FITC Apoptosis Detection Kit (Thermo Fisher, Waltham, MA, USA) according to manufacturers’ protocol. Samples were run directly on a BD Accuri C6 flow cytometer (San Jose, CA, USA).

### 2.7. Metabolic Assays

Oxygen consumption rate (OCR) was measured in Seahorse XF media (Agilent, Santa Clara, CA, USA) supplemented with 100 nM insulin, 1 mM sodium pyruvate, 5.6mM glucose, 4mM glutamine, 1% FCS under basal conditions and in response to 1 μM oligomycin, 1.5 μM FCCP, and 2 μM rotenone + 1 μM Antimycin A using the XFe96 Extracellular Flux Analyzer (Agilent, Santa Clara, CA, USA). Cell-Tak (Corning, Corning, NY, USA) was used for T-cell adherence.

### 2.8. T Cell Transfers and Tumor Model

2.5 × 10^5^ B16F1 melanomas were established subcutaneously on the right flank of female C57BL/6 mice. Tumor-bearing mice were 5Gy irradiated 24 h prior to T cell transfer. After 7 days of tumor growth, 2 × 10^6^ Vehicle or E64FC26-treated Pmel T cells were infused via tail vein to melanoma-bearing mice. Tumor growth was measured every other day with calipers and survival was monitored with an experimental endpoint of tumor growth ≥200 mm^2^.

### 2.9. Statistical Analysis

For data generated using UPR gene arrays, we utilized the ThermoFisher Applied Biosystems Relative Quantitation Analysis Module with applied parameters of corrected *p* value <0.05 and fold-change boundary of 2.0 considered to determine significant differences in gene expression. Tumor growth is analyzed by linear regression of growth curves of vehicle versus drug-treated T cells. Survival to 30 days or tumor size of 200 mm^2^ with Log-rank test for survival proportions of mice treated with vehicle versus E64FC26-treated T cells was used for analysis. Data are presented as standard error of the mean, SEM. Unless otherwise noted, significance was assessed by student’s t-tests. No data were excluded from the analyses. Statistical analyses were performed with GraphPad Prism (Version 8, San Diego, CA, USA) and differences were considered significant when * *p* < 0.05, ** *p* < 0.01, *** *p* <0.001, **** *p* < 0.0001.

## 3. Results

### 3.1. PDI Inhibition Promotes Viability in Healthy T Cells

Targeting PDI is a fruitful strategy to reduce tumor cell viability and control tumor growth [7,23]. The pan-PDI inhibitor E64FC26 was recently identified as an early drug candidate with anti-myeloma activity in vitro and in vivo, with the ability to synergistically enhance the activity of FDA-approved proteasome inhibitors [8]. Targeting redox-dependent proteins is a strategy to enhance T cell tumor control, and compounds that simultaneously boost T cell anti-tumor potential while restricting tumor growth are exciting candidates for cancer immunotherapy. We recently found that repression of ERO1α produced potent anti-tumor immunity of healthy CD8 T cells [6]. Given that ERO1α partners with PDI to carry out redox reactions in the ER lumen, we hypothesized that the newly discovered PDI inhibitor E64FC26 may shape T cell tumor control. We activated Pmel T cells with cognate antigen gp100 and assessed CD8 T cell viability after 3 days of activation in the presence of vehicle or E64FC26 followed by 4 days of ex vivo expansion in the presence of fresh drug. E64FC26 increased CD8 T cell viability, evidenced by the percentage of live T cells (Appendix A, Figure 1A) and reduced Annexin/propidium iodide (PI) positive T cells relative to vehicle controls (Figure 1B). We conducted the study with 0.5 μM E64FC26 given the enhanced T cell viability and previous reports of impaired malignant cell survival at this dose [8].

Inhibiting the isomerase activity of PDI leads to an accumulation of misfolded proteins, activation of the UPR, and apoptosis [7,10]. Surprisingly, the ER stress response in E64FC26-treated healthy CD8 T cells was substantially reduced and supports the enhanced viability shown in Figure 1A,B. Specifically, there was a decrease in *Atf4* and *Ddit3* gene expression (Figure 1C,D). This finding was consistent at the protein level (Figure 1E). However, treatment with E64FC26 only modestly increased the abundance of misfolded proteins in healthy T cells, evidenced by enhanced ubiquitinated proteins in E64FC26-treated CD8 T cells relative to vehicle controls. These data also demonstrate the finding that effector T cells express a high basal level of ubiquitinated proteins (Figure 1E).

Our data prompted us to measure the effect of E64FC26 on oncogenic T cells for the sake of comparison. We found that malignant human T cell lines Hut78 and Jurkat, derived from cutaneous T cell lymphoma and acute T cell leukemia, respectively, showed reduced viability in response to E64FC26 treatment (Figure 1F,G). In contrast to healthy CD8 T cells, E64FC26 induced the UPR in Hut78 and Jurkat T cells (Figure 1H–J). Along these lines, ubiquitination in oncogenic T cell lines increased significantly following E64FC26 treatment relative to vehicle-matched controls (Figure 1J). Together, our data indicate a divergent effect of PDI inhibition on normal versus leukemic T cells, as E64FC26 promotes viability and suppresses UPR in healthy CD8 T cells, but induces UPR and cell death in human T cell lymphoma and leukemia cell lines.

### 3.2. IL-15 Primed T Cells Express Diminished Terminal UPR and Oxidative Stress

The terminal UPR is not well studied in T cell biology. CD8 memory cells are a subset of T cells endowed with traits that increase the capacity of T cells to control tumor growth [24,25]. *In vitro* treatment of CD8 T cells with the cytokine IL-15 generates a pool of memory-like cells that can be studied to understand lineage traits in T cell biology [26,27]. We made the surprising finding that healthy CD8 T cells express a substantial burden of unfolded/misfolded proteins demonstrated by an abundance of ubiquitinated proteins (Figure 1). These data spurred us to measure the global UPR in IL-2 and IL-15-primed T cells to gain insight into regulation of the UPR between IL-2 effector and IL-15 memory-like T cells. A UPR gene array comprised of 83 common UPR genes revealed that the terminal UPR initiated by major ER stress sensor PERK was the crucial UPR axis differentially regulated between IL-2 and IL-15-primed T cells with increased expression in effectors (Figure 2A). Annexin-PI co-staining confirmed reduced cell death among IL-15-primed T cells relative to effectors (Figure 2B).

We previously reported that the enzyme ERO1α was limited in IL-15-primed cells relative to effectors [6] and ERO1α is a transcriptional target of ATF4/CHOP [28]. Consistent with UPR gene expression data, the total ERO1α protein level was limited in IL-15-primed T cells relative to effectors. Both the oxidized and reduced forms ERO1α were diminished. Similarly, total PDI levels were decreased in IL-15-primed cells relative to effector T cells (Figure 2C). Excessive ERO1α activity is a notorious mechanisms of oxidative cell stress that impairs cell survival [29,30,31]. We found that IL-15-primed T cells were limited in global oxidative stress and generation of superoxide in mitochondria, evidenced by reduced DFCDA and MitoSOX fluorescent probe staining, respectively (Figure 2D,E).

### 3.3. PDI Inhibition Diminishes Oxidative Stress in Healthy T Cells

Data in Figure 2 prompted us to assess the relationship of ERO1α and PDI expression in E64FC26-treated T cells. We found that pharmacologic inhibition of PDI resulted in diminished global expression of both oxidized and reduced forms of ERO1α and PDI relative to vehicle controls (Figure 3A). Along these lines, E64FC26 treatment of healthy CD8 T cells decreased global and mitochondrial-associated ROS measured by DCFDA and MitoSOX fluorescent probes, respectively (Figure 3B,C). To confirm that ERO1α was a source of oxidative cell stress in healthy T cells, we activated Pmel T cells with cognate antigen gp100 and expanded T cells in the presence of ERO1α inhibitor EN460 [32]. Assessment of global and mitochondrial ROS in EN460-treated T cells demonstrated that ERO1α contributes to overall oxidative stress and mitochondrial ROS generation in healthy T cells (Figure 3D,E). EN460 treatment of healthy T cells increased CD8 T cell viability evidenced by Annexin-PI co-staining (Figure 3F). Together, our data indicate that PDI-ERO1α redox coupling to maintain isomerase activity in the ER of healthy T cells contributes to generation of oxidative stress that compromises cell viability.

### 3.4. PDI Inhibition Promotes Traits Associated with T cell Tumor Control

We noted overlap in reduced UPR, oxidative stress, and cell death in IL-15 primed memory-like cells and E64FC26-treated T cells (Figure 2 and Figure 3). These data spurred us to measure memory T cell traits that are hallmarks of T cells primed for anti-tumor immunity in E64FC26-treated T cells. FACS phenotyping found that E64FC26-treated cells express a memory T cell phenotype, evidenced by enhanced CD62L surface expression, and diminished activation, measured by CD44 expression (Figure 4A). Next, we assessed expression of hallmark memory T cell stemness gene, transcription factor 7 (*Tcf7*), that has been associated with a superior ability of T cells to control tumor growth [33,34]. We found that E64FC26-treated T cells express *Tcf7* on par with expression found in IL-15-primed T cells (Figure 4B).

T cell metabolism is an integral component that defines T cell anti-tumor capability [35,36]. T cells with sole glycolytic dependence cannot persist in tumors [37], but T cells dependent on the alternate energy source of β-oxidation of fatty acids exhibit superior function in vivo [38,39]. Assessment of gene expression indicated that IL-15 and E64FC26-treated T cells express reduced glycolytic enzyme hexokinase 2 (*Hk2*) and elevate expression of crucial β-oxidation of fatty acid mitochondrial enzyme carnitine palmitoyltransferase 1 (*Cpt1a*) (Figure 4C,D). In line with these data, E64FC26-treated T cells exhibited greater mitochondrial-associated ATP production, maximal respiration, and spare respiratory capacity relative to vehicle controls (Figure 4E,H). To corroborate these data, we performed Western blotting. In accordance with Figure 4B, TCF7 protein was increased in E64FC26-treated T cells relative to vehicle controls (Appendix A). We were unable to detect CPT1A protein, but phosphorylated-Acetyl–CoA Carboxylase (p-ACC) was increased in E64FC26-treated T cells relative to vehicle controls (Appendix A). Phosphorylation of ACC renders the enzyme inactive and blocks inhibition of CPT1A, thereby promoting β-oxidation of fatty acids [40]. Our data indicate that PDI inhibited T cells bear hallmarks of superior T cell anti-tumor metabolism [38,39]. We noted a disconnect between *Hk2* gene and protein expression as HK2 protein was increased in E64FC26-treated T cells (Appendix A). We measured extracellular acidification rate (ECAR) in E64FC26-treated T cells as a functional measure of T cell glycolysis. Basal ECAR was greater in E64FC26-treated T cells, but differences in glycolytic capacity were not detected (Appendix A).

### 3.5. T Cells Treated with PDI Inhibition Promote Superior Tumor Control

Our data indicate that EC64FC26-primed T cells bear phenotypic similarities to IL-15-primed T cells. Given that IL-15-associated T cells harbor superior tumor control relative to effector T cells [38,41,42], we measured the impact of E64FC26-treatment on T cell tumor control. We activated Pmel T cells with cognate antigen gp100 and expanded cells in the presence of vehicle or E64FC26, then transferred T cell groups into mice bearing B16 melanomas. We found that E64FC26-treated T cells controlled tumor growth more effectively than vehicle controls (Figure 5A), leading to increased overall survival in tumor-bearing mice that received E64FC26-conditioned T cells (Figure 5B). These data indicate that conditioning T cells during ex vivo activation and expansion with the PDI inhibitor E64FC26 enhances T cell tumor control in a cellular therapy mouse model. Together, our findings suggest that E64FC26 is a promising oncology drug candidate, with direct anti-tumor activity and the previously uncharacterized potential to bolster healthy T cells’ anti-tumor capability.

## 4. Discussion

These data show a contrasting outcome when pharmacologically modulating PDI in malignant and healthy T cells. Though both cell types increase a misfolded/unfolded protein burden, evidenced by increased ubiquitination in response to PDI inhibition, the UPR is increased in malignant T cells and diminished in healthy T cells (Figure 1). It follows that malignant T cells experience increased death in response to PDI inhibition, but viability in healthy T cells is increased. These data show a dichotomy in response to PDI inhibition between malignant and normal T cells, which is consistent with previous studies showing contrasting effects in normal versus cancer cells [8]. Furthermore, these findings offer PDI inhibition as a potential “Achilles heel” of tumorigenic T cells in addition to the opportunity to enhance normal T cell function for cancer immunotherapy.

The specific mechanism by which PDI inhibition improves survival in healthy T cells remains unclear. Our data suggest that redox crosstalk between PDI and ERO1α is affected by PDI inhibition via global down regulation of both enzymes in healthy T cells (Figure 3). While the PDI-ERO1α relay is essential for redox homeostasis and proper protein folding in the ER lumen [43], in response to severe stress, pathological levels of ERO1α can induce cell death through generation of ROS [28,30,44]. Our data indicate that the mechanism of cell death in effector T cells is predicated by dysregulation of the PDI-ERO1α relay that results in global oxidative cell stress and ROS generation in mitochondria of healthy T cells (Figure 3). The ER is an oxidizing environment buffered by a supply of glutathione [19]. Based on enhanced oxidative stress, our data indicate that excessive PDI-ERO1α activity may undermine glutathione homeostasis in effector T cells. These data are important in the context of cancer immunotherapy, because maintenance of the reduced glutathione pool is essential for T cell tumor control [45] and inflammatory responses [46,47]. The specific role of PDI-ERO1α relay to shape glutathione homeostasis in T cells is an area of interest.

## Figures and Tables

**Figure 1 cells-08-01514-f001:**
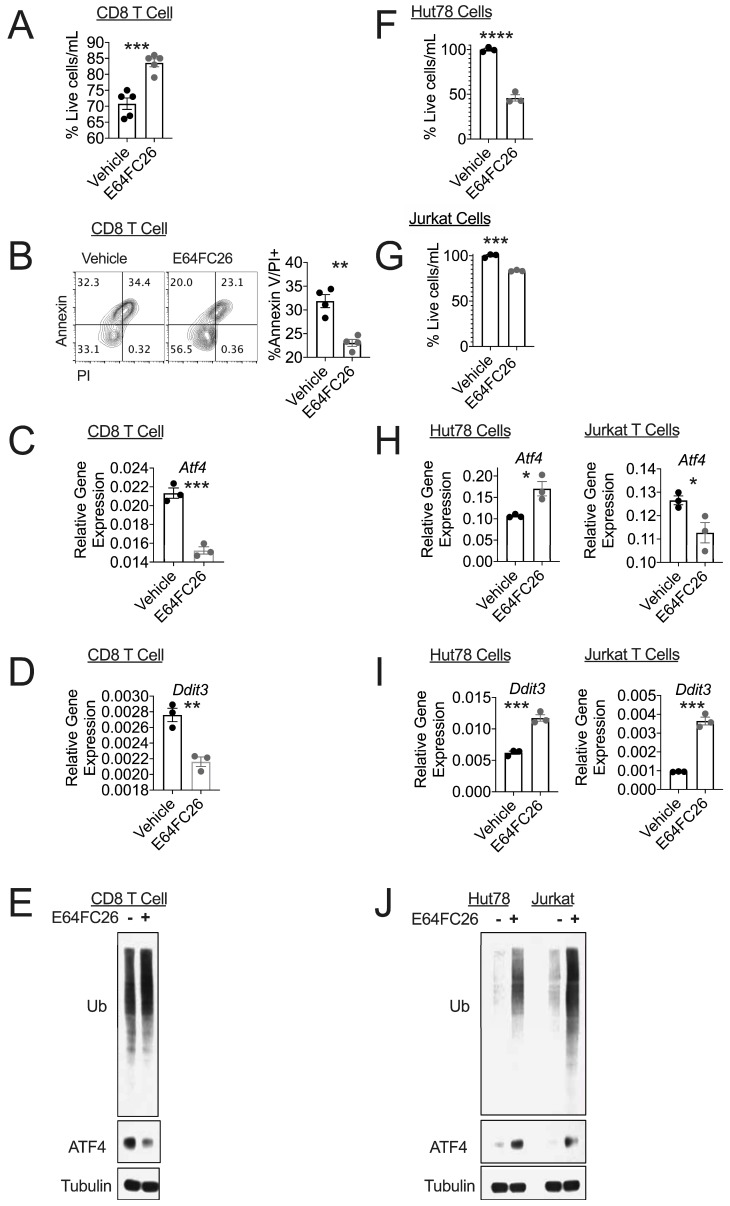
PDI inhibition promotes viability in healthy T cells. Pmel T cells were activated with gp100 peptide and expanded in the presence of vehicle or PDI inhibitor E64FC26. (**A**) Scatter plot with bar graph of percent viable T cells and (**B**) Representative FACS plots and quantification of Annexin V expression co-stained with propidium iodide (PI) and (**C–D**) Scatter plot with bar graphs of RT-PCR used to measure expression of indicated genes and (**E**) immunoblot for indicated proteins with Tubulin as loading control. Densitometry quantification normalized to Tubulin; Ubiquitin: Vehicle = 0.69, E64FC26 = 0.82, ATF4: Vehicle = 0.68, EC64FC26 = 0.29. Data points represent combined values from three individual experiments. Immunoblot repeated twice. Hut78 and Jurkat T cells were treated for 16 h with vehicle or protein disulfide isomerase (PDI) inhibitor E64FC26. Scatter plot with bar graph of percent viable T cells in (**F**) Hut78 and (**G**) Jurkat T cells is shown. (**H–I**) Scatter plot with bar graphs of RT-PCR used to measure expression of indicated genes and (**J**) immunoblot for indicated proteins with Tubulin as loading control. Densitometry quantification normalized to Tubulin; Hut78: Ubiquitin: Vehicle = 0.05, E64FC26 = 0.54, ATF4: Vehicle = 0.06, EC64FC26 = 0.57, Jurkat: Ubiquitin: Vehicle = 0.16, E64FC26 = 0.99, ATF4: Vehicle = 0.01, EC64FC26 = 0.48. Data points represent combined values from individual experiments. Immunoblot repeated twice. Differences were considered significant when * *p* < 0.05, ** *p* < 0.01, *** *p* < 0.001, **** *p* < 0.0001.

**Figure 2 cells-08-01514-f002:**
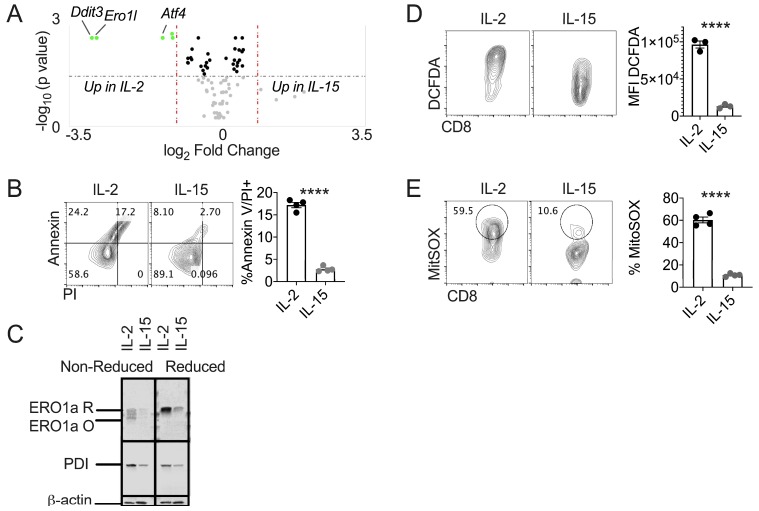
IL-15 primed T cells express reduced terminal UPR and oxidative stress. Pmel T cells were activated with gp100 peptide and expanded for 3 days then differentiated with IL-2 or IL-15. (**A**) Volcano plot of gene array data of 83 common UPR genes, differentially expressed genes in the terminal UPR pathway are annotated. Arrays were run in triplicate of three biological replicates in each condition. (**B**) Representative FACS plots and quantification of Annexin V expression co-stained with PI (**C**) Immunoblot to detect oxidized and reduced forms of indicated proteins with total protein as control. Densitometry quantification normalized to β-actin; Non-reduced: ERO1: IL-2 = 0.51, IL-15 = 0.10, PDI: IL-2 = 0.79, IL-15 = 0.21, Reduced: ERO1: IL-2 = 1.13, IL-15 = 0.25, PDI: IL-2 = 0.61, IL-15 = 0.17. (**D**) Representative FACS plots and quantification of (**E**) expression of global oxidative cell stress with DCFDA indicator or (**F**) expression of mitochondrial superoxide with MitoSOX indicator. Data points represent replicates within a given experiment. Individual experiments repeated three times. Differences were considered significant when **** p < 0.0001.

**Figure 3 cells-08-01514-f003:**
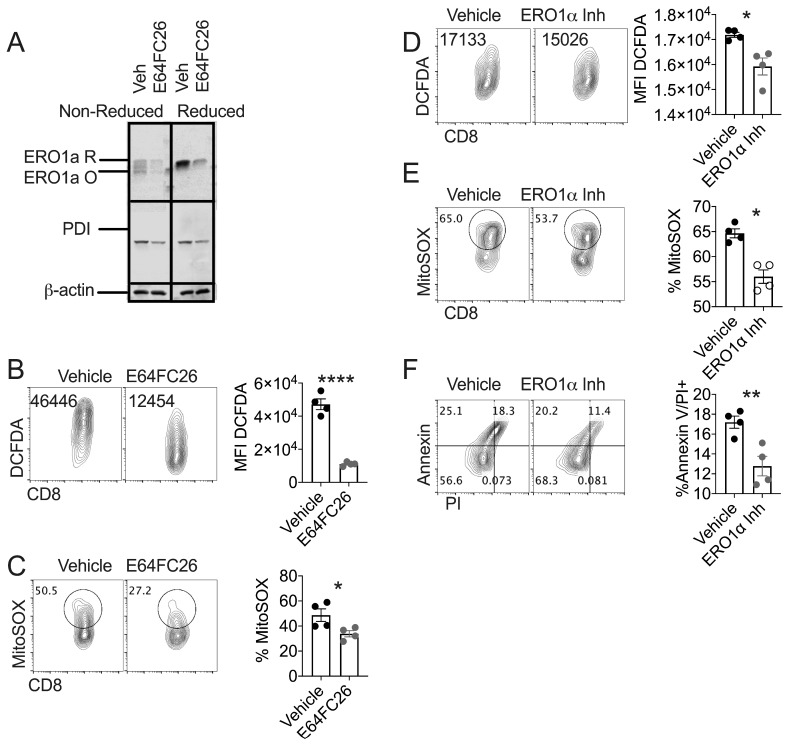
PDI inhibition diminishes oxidative stress in T cells. Pmel T cells were activated with gp100 peptide and expanded in the presence of vehicle or PDI inhibitor E64FC26. (**A**) Immunoblot to detect oxidized and reduced forms of PDI and ERO1α with total protein as control. Densitometry quantification normalized to β-actin; Non-reduced: ERO1: IL-2 = 0.42, IL-15 = 0.16, PDI: IL-2 = 0.42, IL-15 = 0.23, Reduced: ERO1: IL-2 = 0.79, IL-15 = 0.22, PDI: IL-2 = 0.40, IL-15 = 0.24. Representative FACS plots and quantification of (**B**) global oxidative cell stress with DCFDA indicator or (**C**) mitochondrial superoxide with MitoSOX indicator. Pmel T cells were activated with gp100 peptide and expanded in the presence of vehicle or ERO1α inhibitor EN460. Representative FACS plots and quantification of (**D**) global oxidative cell stress with DCFDA indicator or (**E**) mitochondrial superoxide with MitoSOX indicator. (**F**) Representative FACS plots and quantification of Annexin V expression co-stained with PI. Data points represent replicates within a given experiment. Individual experiments repeated three times. Differences were considered significant when * *p* < 0.05, ** *p* < 0.01, **** *p* < 0.0001.

**Figure 4 cells-08-01514-f004:**
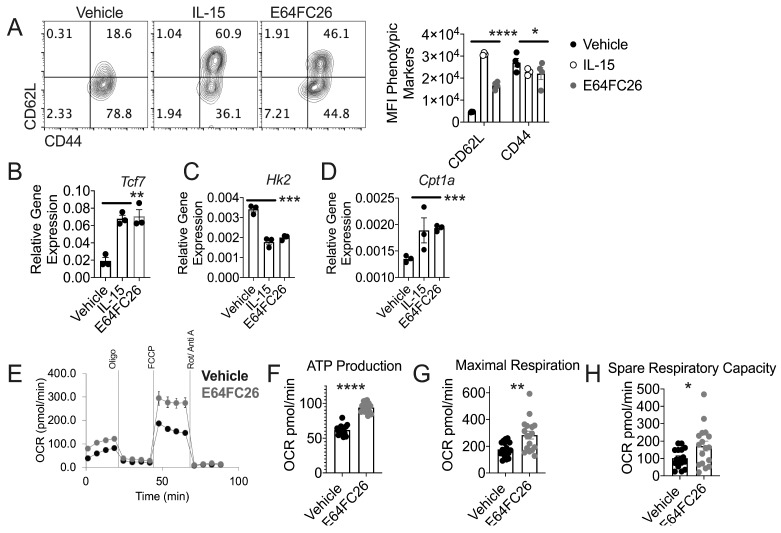
PDI inhibition promotes traits associated with T cell tumor control. Pmel T cells were activated with gp100 peptide and expanded in the presence of vehicle, IL-15, or E64FC26. (**A**) Representative FACS plots and quantification of CD62L and CD44 expression and (**B**–**D**) RT-PCR was used to measure expression of indicated genes. Data points represent average values from three separate experiments. Significance represents vehicle versus E64FC26 values, IL-15-primed T cells values are shown as a reference point. (**E**) Oxygen consumption rate (OCR) trace with injection markers and (**F**–**H**) quantification of indicated values from Seahorse Bioanalysis in response to mitochondrial stress test. Individual experiment performed three times. Differences were considered significant when * *p* < 0.05, ** *p* < 0.01, *** *p* <0.001, **** *p* < 0.0001.

**Figure 5 cells-08-01514-f005:**
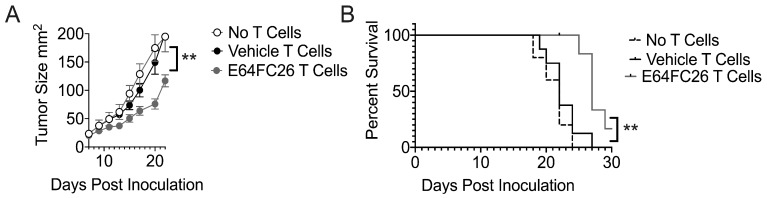
T cells treated with PDI inhibition promote superior tumor control. Mice bearing 7-day established B16 melanomas were intravenously infused with 2 × 10^6^ Pmel T cells activated with gp100 peptide and expanded in the presence of vehicle or PDI inhibitor E64FC26. (**A**) Tumor growth was monitored every other day with calipers and (**B**) time to sacrifice was recorded. Linear regression and Log-rank test for survival proportions of mice were used to test significant differences in tumor growth and survival, respectively, between vehicle and EG64FC2 T cell infused groups. Four (no T cells), seven, seven mice per group. Individual experiment performed twice. Differences were considered significant when ** *p* < 0.01.

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
