# Peer review of "Endoplasmic Reticulum Protein Disulfide Isomerase Shapes T Cell Efficacy for Adoptive Cellular Therapy of Tumors"

_cells, 2019, doi:10.3390/cells8121514_

Round 1
Reviewer 1 Report
This manuscript by Hurst et al. examined the effect of protein disulfide isomerase (PDI) inhibitor E64FC26 on UPR in normal T cells. They observed that E64FC26 increased cell survival and decreased UPR in normal T cells, whereas opposite effect was observed in oncogenic T cell lines. The E64FC26 treated cells had similar effect to the IL-15 primed memory T-cells. Finally, E64FC26 treated cells showed significant reduction in melanoma cell growth in vivo.
There are some concerns which needs to be addressed:
Only one concentration of E64FC26 was used. Incubation time is not indicated. It will be important to see dose and incubation time course for E64FC26. Also, no negative and positive controls are included. Please quantify blots in Fig1E, 1J, 2C and 3A. In fig2C and 3A, it will be important to include a non-redox-responsive protein as a control. Fig.4B, it will be important to also examine the protein levels of these genes, as often gene and protein levels do not match. Fig. 4E, all controls are used, but they are not clearlt depicted the figure where particular inhibitors are added. It will also be interesting to look at the effect of E64FC26 on glycolysis, via measuring extracellular acid release using seahorse analyzer. Fig5, it will be important to compare the effect of E64FC26 with IL-15 priming.
Minor concern: x-axis for Fig5B, "Days" is missing.
Reviewer 2 Report
this manuscript details a logical series of experiments on PD1 inhibitor E64FC26 and T cells showing that the inhibitor promotes viability in CD8 cells and restricts the survival of oncogenic T cells. this suggests the PD1 inhibitor may be used some day to treat cancer patients to induce tumor cell death and simultaneously nourish anti-tumor immunity. the data are presented clearly and logically.
my one objection is that the Mycoplasm test was performed in 2017 which seems too long ago.
Round 2
Reviewer 1 Report
Authors have done the required changes and this manuscript can be accepted now.